# FLOWMS: FLOW MATCHING FOR DE NOVO STRUCTURE ELUCIDATION FROM MASS SPECTRA

**Jianan Nie**
Virginia Tech
jianan@vt.edu

**Peng Gao**
Virginia Tech
penggao@vt.edu

## ABSTRACT

Mass spectrometry (MS) stands as a cornerstone analytical technique for molecular identification, yet *de novo* structure elucidation from spectra remains challenging due to the combinatorial complexity of chemical space and the inherent ambiguity of spectral fragmentation patterns. Recent deep learning approaches, including autoregressive sequence models, scaffold-based methods, and graph diffusion models, have made progress. However, diffusion-based generation for this task remains computationally demanding. Meanwhile, discrete flow matching, which has shown strong performance for graph generation, has not yet been explored for spectrum-conditioned structure elucidation. In this work, we introduce FlowMS, the first discrete flow matching framework for spectrum-conditioned *de novo* molecular generation. FlowMS generates molecular graphs through iterative refinement in probability space, enforcing chemical formula constraints while conditioning on spectral embeddings from a pretrained formula transformer encoder. Notably, it achieves state-of-the-art performance on 5 out of 6 metrics on the NPLIB1 benchmark: 9.15% top-1 accuracy (9.7% relative improvement over DiffMS) and 7.96 top-10 MCES (4.2% improvement over MS-BART). We also visualize the generated molecules, which further demonstrate that FlowMS produces structurally plausible candidates closely resembling ground truth structures. These results establish discrete flow matching as a promising paradigm for mass spectrometry-based structure elucidation in metabolomics and natural product discovery.

## 1 INTRODUCTION

Mass spectrometry (MS) is a foundational analytical technique in chemistry and biology, enabling the identification and characterization of small molecules across diverse applications, including drug discovery (Aebersold & Mann, 2003), metabolomics (Gentry et al., 2024), and natural product identification (Atanasov et al., 2021). In MS analysis, molecules are fragmented into charged ions, producing characteristic mass-to-charge ratio patterns that serve as structural fingerprints for retrieval and compound identification. However, such retrieval-based methods require extensive annotated spectral databases and cannot recover missing compound structures (Bittremieux et al., 2022). Alternatively, *de novo* structure elucidation offers the potential to discover new compounds.

Nevertheless, *de novo* structure elucidation or inverse MS problem, determining molecular structure from mass spectra, poses fundamental challenges. From a computational perspective, reconstructing chemical structures from fragment masses is combinatorially complex, as the space of possible structures explodes with molecular size (Bohde et al., 2025). From a chemical perspective, mass spectra are inherently underspecified: multiple distinct structures can produce nearly identical fragmentation patterns, as exemplified by structural isomers such as leucine and isoleucine, which differ only in side-chain branching (Brown et al., 1997). To address these problems, artificial intelligence models can be trained on large-scale spectral datasets to generate structurally plausible candidate molecules, substantially narrowing the search space for further experimental validation.

Recent advances in deep generative models have shown promise for *de novo* structure elucidation from mass spectra. Fingerprint-based approaches (Stravs et al., 2022; Goldman et al., 2023b) leverage intermediate representations to enable pretraining on larger molecular datasets, while scaffold-based methods (Wang et al., 2025) decompose generation into core structure prediction followed

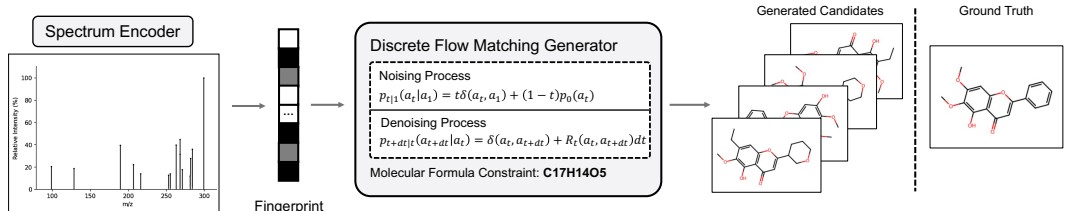

Figure 1: Overview of our discrete flow matching framework for mass spectrum-guided molecular generation. Given a tandem mass spectrum and molecular formula, the spectrum encoder produces a conditioning fingerprint. The discrete flow matching generator produces candidate molecular structures, which are subsequently ranked by spectral frequency.

by completion. Meanwhile, MS-BART (Han et al., 2025) treats structure generation as sequence-to-sequence translation, leveraging transformer architectures with fingerprint-based pretraining. Recently, DiffMS (Bohde et al., 2025) introduced discrete graph diffusion for this task, demonstrating outstanding performance by handling formula constraints and the one-to-many mapping from spectra to structures. While these approaches achieve strong performance, whether alternative generative frameworks can further improve structure elucidation quality remains an open question.

*Discrete flow matching* (Lipman et al., 2022; Liu et al., 2022; Gat et al., 2024) has emerged as a compelling alternative to diffusion models for generative tasks. Unlike discrete diffusion, which couples its forward corruption schedule to the reverse generative process, discrete flow matching employs explicit linear interpolation paths between noise and data distributions (Gat et al., 2024), decoupling training from sampling and enabling flexible inference selection. Specifically, Qin et al. (2024) demonstrates the effectiveness of flow matching for molecular graph generation, achieving competitive performance on standard benchmarks. However, adapting discrete flow matching to *de novo* structure elucidation from mass spectra remains unexplored with unique challenges: the model must learn to condition on complex mass spectral embeddings while handling the inherent ambiguity where multiple valid structures produce similar fragmentation patterns.

In this work, we introduce **FlowMS**, the discrete flow matching framework for *de novo* molecular generation from mass spectra. FlowMS adapts the discrete flow matching formulation under the spectrum-conditioned setting, employing linear interpolation noising and continuous-time Markov chain (CTMC) denoising while enforcing chemical formula constraints throughout generation. We pair this decoder with the MIST formula transformer encoder (Goldman et al., 2023a) and follow an encoder-decoder pretraining strategy to leverage large-scale fingerprint-molecule pairs. Experiments on the widely adopted NPLIB1 benchmark demonstrate that FlowMS achieves state-of-the-art performance on 5 out of 6 evaluation metrics, including all top-1 metrics and top-10 structural similarity. Specifically, FlowMS attains 9.15% top-1 accuracy, surpassing the previous best of 8.34% (DiffMS), while also improving structural similarity over the strongest baseline MS-BART (top-1 MCES: 9.32 vs. 9.66; top-1 Tanimoto: 0.46 vs. 0.44). We additionally visualize generated molecules alongside ground truth structures to examine their structural correspondence. These results establish discrete flow matching as a promising paradigm for mass spectrometry-based structure elucidation.

## 2 RELATED WORK

**Mass Spectra Modeling.** Tandem mass spectrometry (MS/MS) generates fragment spectra as collections of mass-to-charge (m/z) peaks with corresponding intensities, forming variable-length, two-dimensional data that poses distinct modeling challenges. A straightforward encoding strategy involves spectral binning, which discretizes the continuous m/z space into fixed intervals (e.g., 0.1 or 0.01 Da bins) (Seddiki et al., 2020), producing fixed-size input vectors amenable to convolutional architectures. Methods such as Spec2Mol (Litsa et al., 2023) and MS2DeepScore (Huber et al., 2021b) employ this binning approach with 1D CNNs or Siamese networks to learn spectral embeddings. However, binned representations suffer from sparsity and sensitivity to peak alignment errors, limiting their ability to capture fine-grained spectral patterns. To address these limitations,

some approaches leverage molecular fingerprints as semantically richer intermediate representations. CSI:FingerID (Dührkop et al., 2015) introduces this direction by predicting molecular fingerprints from tandem mass spectra using fragmentation trees and support vector machines, enabling effective structure ranking from chemical databases. MSNovelist (Stravs et al., 2022) builds on this by integrating predicted fingerprints into an encoder-decoder model for de novo structure generation. Furthermore, MIST (Goldman et al., 2023a) employs transformer-based architectures to predict fingerprints directly from peak lists. Building on these approaches, our method utilizes molecular fingerprints as an intermediate spectral representation. This strategy enables scalable pretraining while ensuring that the learned embeddings effectively preserve essential chemical semantics.

**Structure Elucidation from Mass Spectrometry.** Molecular structure elucidation from mass spectrometry is primarily characterized by two paradigms, including spectral library matching and *de novo* molecular generation. Spectral library matching formulates structure elucidation as an information retrieval problem, comparing query spectra against curated databases of experimental or simulated spectrum data (Wang et al., 2016). For instance, Spec2Vec (Huber et al., 2021a) utilizes learned spectral embeddings to facilitate retrieval over databases like GNPS (Wang et al., 2016), while CFM-ID (Wang et al., 2021) addresses the scarcity of experimental data by simulating spectra from comprehensive molecular databases such as PubChem (Kim et al., 2016; 2019; 2023). While effective for known chemical compounds, these approaches are inherently constrained by the scope of reference libraries and the quality of mass spectra, restricting their capacity to identify truly novel molecules. Conversely, *de novo* generation methods bypass these limitations by directly generating molecular structures from mass spectral data, thereby facilitating the exploration of novel molecules within the uncharted chemical space. In this domain, Spec2Mol (Litsa et al., 2023) employs a CNN encoder to map spectra into a latent space coupled with an autoregressive SMILES decoder, while MSNovelist (Stravs et al., 2022) generates structures from fingerprints predicted by CSI:FingerID using an LSTM-based sequence model. More recently, MADGEN (Wang et al., 2025) proposes a two-stage approach that retrieves molecular scaffolds and then generates scaffold-conditioned full structures. MS-BART (Han et al., 2025) bridges spectral and molecular modalities through a unified token vocabulary, by large-scale pretraining on fingerprint molecule pairs. DiffMS (Bohde et al., 2025) adopts a discrete graph diffusion framework to generate molecules conditioned on spectral embeddings from MIST (Goldman et al., 2023a) and molecular formula constraints.

**Generative Models for Molecular Design.** Generative models have become essential in molecular generation due to their ability to approximate complex distributions in chemical space (Jin et al., 2018; De Cao & Kipf, 2018; Gebauer et al., 2019; Shi et al., 2020; Hoogeboom et al., 2022; Peng et al., 2023; Liu et al., 2024; 2025). Among these, diffusion models have emerged as powerful frameworks for molecular generation, building on success in image (Ho et al., 2020; Song et al., 2020; Bar-Tal et al., 2023) and text domains (Austin et al., 2021; Li et al., 2022). For graph-structured molecular data, DiGress (Vignac et al., 2022) introduced discrete graph diffusion through categorical noise processes on node and edge features, enabling non-autoregressive generation with permutation invariance. Flow matching (Lipman et al., 2022; Liu et al., 2022; Gat et al., 2024) has emerged as a compelling alternative to diffusion models, learning continuous-time transport processes with more stable training dynamics and sampling through deterministic ODE integration. DeFoG (Qin et al., 2024) extends flow matching to discrete graph generation by constructing trajectories in probability space and achieving competitive performance.

While these advances establish flow matching as an effective framework for vision, language, and graph generation, its application to spectrum-conditioned structure elucidation remains unexplored. This setting presents unique challenges: the model must condition on complex spectral embeddings while strictly enforcing chemical formula constraints, an important inductive bias from mass spectrometry that defines the target atom composition. In this work, we adapt discrete flow matching (Qin et al., 2024) to this conditional generation task, demonstrating that flow matching provides effective probability transport for spectrum-to-structure prediction while accommodating formula constraints.

## 3 METHODOLOGY

### 3.1 PROBLEM FORMULATION

We formulate the structure elucidation task as conditional molecular generation from mass spectrometry data. A training instance consists of a structure-spectrum pair $(\mathcal{M}, \mathcal{S})$, where $\mathcal{M}$ represents the

molecular graph and $\mathcal{S}$ denotes the corresponding tandem mass spectrum. Our *objective* is to generate a ranked list of $k$ candidate molecules $\{\widehat{\mathcal{M}}_1, \ldots, \widehat{\mathcal{M}}_k\}$ that best explain the observed spectrum $\mathcal{S}$. We represent a molecular graph as $\mathcal{M} = (\mathbf{A}, \mathbf{X}, \mathbf{y})$, where $\mathbf{A} \in \{0,1\}^{n \times n \times b}$ is a one-hot encoded adjacency tensor representing bonds between $n$ heavy atoms, and $b$ represents the number of bond types. We use $b = 5$, including no bond, single, double, triple, and aromatic bonds. $\mathbf{X} \in \mathbb{R}^{n \times d}$ denotes node features, such as atom types derived from the molecular formula. $\mathbf{y} \in \mathbb{R}^c$ denotes a spectrum-derived conditioning vector encoded from the mass spectrum, where $c$ is the dimension of features. Since atom types are determined by the formula, our generation task reduces to predicting the adjacency structure $\mathbf{A}$ conditioned on fixed node features $\mathbf{X}$ and spectral embedding $\mathbf{y}$.

Notably, an important physical prior in mass spectrometry is the molecular formula, which constrains the molecular search space by determining the number and types of atoms (C, N, O, S, P, etc.). The formula information can be inferred from high-resolution MS1 precursor mass and isotopic patterns using tools such as SIRIUS (Dührkop et al., 2019) and MIST-CF (Goldman et al., 2023b). These tools routinely achieves high accuracy for compounds within typical natural product mass range. Following prior work (Bohde et al., 2025), we model only heavy-atom connectivity and infer hydrogen placement implicitly, as heavy-atom topology captures the essential structural information while reducing computational complexity. So the generated molecules may differ in formula from the true molecule in the hydrogen atom count, but remains the same for the heavy atom count.

## 3.2 DISCRETE FLOW MATCHING FOR MOLECULAR GRAPHS

We now describe the discrete flow matching formulation for molecular graph generation with three core components: the noising process, the denoising process, and the training objective.

**Noising Process.** We adopt the discrete flow matching framework for graph generation, which constructs trajectories in probability simplex space for edge features. For a molecular graph $\mathcal{M} = (\mathbf{A}, \mathbf{X}, \mathbf{y})$ at time $t \in [0, 1]$, we denote the time-dependent state as $\mathcal{M}_t = (\mathbf{A}_t, \mathbf{X}, \mathbf{y})$, where $\mathbf{A}_t$ represents the noisy adjacency matrix at time $t$ while node features $\mathbf{X}$ and conditioning $\mathbf{y}$ remain fixed. The noising process from $t = 1$ (clean data) to $t = 0$ (noise) interpolates linearly from the clean data distribution. Specifically, for each edge $a_{ij}$ in the adjacency matrix, the noising trajectory $p_{t|1}(a_t^{(ij)}|a_1^{(ij)}) \in [0, 1]$ is defined as:

$$p_{t|1}(a_t^{(ij)}|a_1^{(ij)}) = t \cdot \delta(a_t^{(ij)}, a_1^{(ij)}) + (1 - t) \cdot p_0(a_t^{(ij)}), \tag{1}$$

where $\delta(\cdot, \cdot)$ is the Kronecker delta, which equals 1 when the indices are equal and 0 otherwise. $a_1^{(ij)}$ denotes the true bond type. $p_0$ represents the initial distribution over bond types, which is initialized as the uniform distribution over all bond categories. Since molecular graphs are undirected, we apply the noising process only to the upper triangular portion of $\mathbf{A}$ and symmetrize the result. The noising trajectory applies noise to each edge independently:

$$p_{t|1}(\mathbf{A}_t|\mathbf{A}_1) = \prod_{i<j} p_{t|1}(a_t^{(ij)}|a_1^{(ij)}). \tag{2}$$

**Denoising Process.** The denoising process from $t = 0$ (noise) to $t = 1$ (clean data) is governed by a continuous-time Markov chain (CTMC) characterized by a rate matrix $\mathbf{R}_t \in \mathbb{R}^{k \times k}$ that defines instantaneous transition rates of initial distribution in time $t \in [0, 1]$. For each edge, the transition probability over an infinitesimal time step $dt$ is:

$$p_{t+dt|t}(a_{t+dt}^{(ij)}|a_t^{(ij)}) = \delta(a_t^{(ij)}, a_{t+dt}^{(ij)}) + R_t^{(ij)}(a_t^{(ij)}, a_{t+dt}^{(ij)}) \, dt, \tag{3}$$

where $R_t^{(ij)}(a_t^{(ij)}, a_{t+dt}^{(ij)})$ denotes the rate of transitioning from bond type $a_t^{(ij)}$ to $a_{t+dt}^{(ij)}$. The conditional rate matrix for edge $(i, j)$ given the clean bond type $a_1^{(ij)}$ is computed as:

$$R_t^{(ij)}(a_t^{(ij)}, a_{t+dt}^{(ij)}|a_1^{(ij)}) = \frac{\max\{0, \partial_t p_{t|1}(a_{t+dt}^{(ij)}|a_1^{(ij)}) - \partial_t p_{t|1}(a_t^{(ij)}|a_1^{(ij)})\}}{Z_t^{>0} \, p_{t|1}(a_t^{(ij)}|a_1^{(ij)})}, \tag{4}$$

where $Z_t^{>0}$ is the number of bond types with non-zero probability at time $t$.

We start by sampling a purely noisy adjacency matrix $\mathbf{A}_0$ from the predefined initial distribution

$$p_0(\mathcal{M}_0) = \prod_{i<j} p_0\left(a_0^{(ij)}\right),  \tag{5}$$

where edges are assumed to be independently initialized.

During sampling, we employ independent Euler steps in the denoising process, with a finite time step $\Delta t$:

$$\mathbf{A}_{t+\Delta t} \sim \prod_{i<j} \tilde{p}_{t+\Delta t|t}^{(ij)}(a_{t+\Delta t}^{(ij)}|\mathcal{M}_t),  \tag{6}$$

Each term $\tilde{p}_{t+\Delta t|t}^{(ij)}(a_{t+\Delta t}^{(ij)}|\mathcal{M}_t)$ corresponds to the Euler step given in Equation (3), where transition dynamics are governed by the rate matrix:

$$R_t^{(ij)}(a_t^{(ij)}, a_{t+\mathrm{d}t}^{(ij)}) = \mathbb{E}_{p_{1|t}^{(ij)}(a_1^{(ij)}|\mathcal{M}_t)}\left[R_t^{(ij)}(a_t^{(ij)}, a_{t+\mathrm{d}t}^{(ij)}|a_1^{(ij)})\right].  \tag{7}$$

**Training Objective.** The rate matrix used for the denoising process in Equation (7) requires the knowledge of the marginal distribution over bond types for each edge, $p_{1|t}^{(ij)}(a_1^{(ij)}|\mathcal{M}_t)$, which is generally intractable. Thus, we define and train a neural network $f_\theta$ parameterized by $\theta$ to approximate $\boldsymbol{p}_{1|t}^\theta(\mathbf{A}_1|\mathcal{M}_t) = f_\theta(\mathcal{M}_t, t)$ and predict the denoised adjacency matrix given noisy input $\mathcal{M}_t$. Lastly, the training loss is formulated as cross-entropy between predicted and true adjacency matrices, aggregated over time steps $t \sim \mathcal{U}[0,1]$:

$$\mathcal{L} = \mathbb{E}_{t\sim\mathcal{U}[0,1],\, p_1(\mathcal{M}_1),\, p_{t|1}(\mathcal{M}_t|\mathcal{M}_1)}\left[-\sum_{i<j} \log p_{1|t}^{\theta,(ij)}(a_1^{(ij)}|\mathcal{M}_t)\right].  \tag{8}$$

Since node features $\mathbf{X}$ are fixed by the molecular formula constraint, the loss only includes edge predictions, in contrast to the general discrete flow matching formulation (Qin et al., 2024), which also optimizes node type predictions.

## 3.3  MODEL ARCHITECTURE AND PRETRAINING STRATEGY

Our model follows an encoder-decoder architecture that enables independent pretraining of each component before end-to-end finetuning on spectrum-to-molecule generation. The discrete flow matching framework for mass spectrum-guided molecular generation is illustrated in Figure 1.

**Spectrum Encoder.** We adopt the MIST formula transformer (Goldman et al., 2023a) as our spectrum encoder. MIST treats a mass spectrum as a set of $(m/z, \text{intensity})$ peaks. A set transformer with multi-head attention processes these peak embeddings while implicitly modeling neutral loss relationships between fragments. We extract the final hidden state corresponding to the precursor peak as the structural conditioning vector for the graph decoder. To pretrain the encoder, we follow Bohde et al. (2025) to train it to predict 2048-bit Morgan fingerprints from spectra. This pretraining objective encourages the encoder to extract chemically meaningful structural information from fragmentation patterns, providing a strong initialization for subsequent spectrum-to-molecule finetuning.

**Graph Decoder.** The flow matching decoder $f_\theta$ is implemented as a Graph Transformer (Dwivedi & Bresson, 2020) that predicts clean adjacency distributions from noisy graphs. The architecture consists of: (1) separate MLPs to encode edge features $\mathbf{A}_t$, node features $\mathbf{X}$, and conditioning vector $\mathbf{y}$; (2) Graph Transformer layers with multi-head attention and residual connections; and (3) output MLP that predicts per-edge probability distributions over bond types and the denoised adjacency matrix. We pretrain the decoder on a dataset of 2.8 million fingerprint-molecule pairs provided by Bohde et al. (2025), which is sampled from DSSTox (EPA, 2015), HMDB (Wishart et al., 2022), COCONUT (Sorokina et al., 2021), and MOSES (Polykovskiy et al., 2020). During pretraining, we directly use molecular fingerprints as conditioning $\mathbf{y}$ rather than spectrum embeddings such that the decoder is trained to generate molecules subject to structural constraints. We exclude all molecules from the NPLIB1 and MassSpecGym from pretraining data to ensure a setting where the model is generating truly novel structures. After pretraining, we jointly finetune the encoder and decoder on paired spectrum-molecule data. The encoder produces spectral embeddings, which condition on the decoder's flow matching process. This end-to-end training adapts the model to the distribution of mass spectra and their corresponding molecular structures.

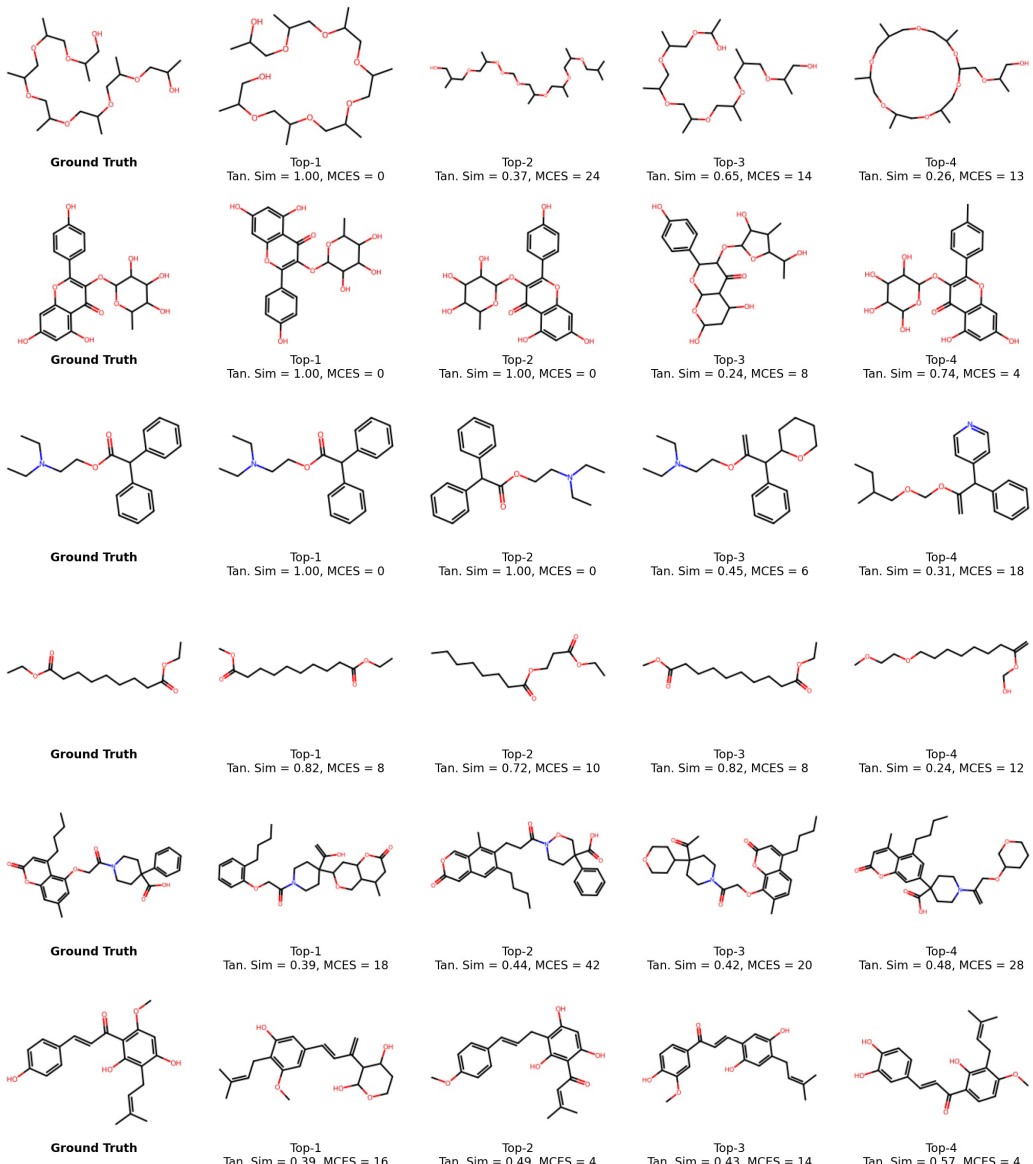

Figure 2: Generated molecules on representative NPLIB1 samples, with the ground truth structure shown in the left column and the FlowMS predictions in the right columns. Tanimoto similarity scores and Maximum Common Edge Substructure (MCES) values are indicated below each prediction.

# 4 EXPERIMENTS

## 4.1 DATASETS AND BASELINES

**Datasets.** We evaluate our approach on NPLIB1 (Dührkop et al., 2021), a widely adopted benchmark for mass spectrometry-based structure elucidation. NPLIB1 is derived from the GNPS spectral library (Wang et al., 2016), originally curated for training the CANOPUS tool (Dührkop et al., 2021), which predicts compound classes, such asalcohols, phenol ethers, and others. Its name serves to distinguish the dataset from the associated tool. NPLIB1 contains high-quality tandem mass spectra paired with verified molecular structures, and has been used recently to benchmark other metabolomics tools such as MIST(Goldman et al., 2023a). The dataset is partitioned into training,

Table 1: *De novo* structural elucidation performance on NPLIB1 (Dührkop et al., 2021) dataset. The best performing model for each metric is **bold** and the second best is underlined. ∗ indicates results reproduced from DiffMS. Methods are approximately ordered by performance.

| Model | Top-1 | | | Top-10 | | |
|---|---|---|---|---|---|---|
| | Accuracy ↑ | MCES ↓ | Tanimoto ↑ | Accuracy ↑ | MCES ↓ | Tanimoto ↑ |
| Spec2Mol∗ | 0.00% | 27.82 | 0.12 | 0.00% | 23.13 | 0.16 |
| MIST + Neuraldecipher∗ | 2.32% | 12.11 | 0.35 | 6.11% | 9.91 | 0.43 |
| MIST + MSNovelist∗ | 5.40% | 14.52 | 0.34 | 11.04% | 10.23 | 0.44 |
| MADGEN | 2.10% | 20.56 | 0.22 | 2.39% | 12.64 | 0.27 |
| DiffMS | 8.34% | 11.95 | 0.35 | **15.44%** | 9.23 | 0.47 |
| MS-BART | 7.45% | 9.66 | 0.44 | 10.99% | 8.31 | **0.51** |
| FlowMS | **9.15%** | **9.32** | **0.46** | 12.05% | **7.96** | **0.51** |

validation, and test sets based on molecular structure similarity, measured by the edit distance on Morgan fingerprints.

**Baselines.** Spec2Mol (Litsa et al., 2023) is retrained on the NPLIB1 dataset to enable fair comparison. MIST+MSNovelist modifies the original MSNovelist framework (Stravs et al., 2022) by replacing CSI:FingerID (Goldman et al., 2023a) with MIST. Similarly, MIST+Neuraldecipher encodes molecules into CDDD representations (Winter et al., 2019), followed by SMILES string reconstruction using a pretrained LSTM decoder. Among recent approaches, MADGEN (Wang et al., 2025) retrieves molecular scaffolds, then generates complete structures using the RetroBridge model (Igashov et al., 2023), conditioned on both spectra and scaffolds. DiffMS (Bohde et al., 2025) utilizes MIST as the spectrum encoder and employs a Graph Transformer for the diffusion decoder, with separate pretraining of encoder and decoder components. MS-BART (Han et al., 2025) adopts a unified token for fingerprints and SELFIES representations, enabling cross-modal pretraining followed by finetuning on MIST-predicted fingerprints from experimental spectra. The MADGEN$_{Oracle}$ in Wang et al. (2025) feeds in the ground truth scaffold, and MS-BART(Gold Fingerprint) in Han et al. (2025) feeds in the ground truth fingerprints, both are strong structural priors that do not fall within the setting of complete *de novo* generation, and are thus not included in our evaluation.

## 4.2 EVALUATION METRICS AND RANKING STRATEGIES

**Evaluation Metrics.** We adopt the *de novo* generation metrics established by Bushuiev et al. (2024), consistent with recent works in mass spectrum-to-molecule generation (Bohde et al., 2025; Han et al., 2025; Wang et al., 2025). Following standard practice, we generate 100 candidate molecules per spectrum and evaluate the following metrics for top-$k$ predictions:

- **Top-$k$ accuracy:** Measures whether the ground truth molecule appears in the top-$k$ predictions via exact InChIKey matching.
- **Top-$k$ maximum Tanimoto similarity:** Computes the structural similarity of the closest molecule to the ground truth within the top-$k$ predictions using 2048-bit Morgan fingerprints (Morgan, 1965) with radius 2.
- **Top-$k$ minimum MCES:** Calculates the graph edit distance between the closest predicted molecule and the ground truth using the maximum common edge subgraph metric (Kretschmer et al., 2023), where MCES = 0 indicates identical molecular graphs.

Following previous works (Bohde et al., 2025; Bushuiev et al., 2024; Wang et al., 2025; Han et al., 2025), we report the $k = 1, 10$ in this work.

**Candidate Ranking.** To obtain a ranked list of predictions, we follow the methodology established by DiffMS (Bohde et al., 2025): we sample 100 molecules for each spectrum, remove invalid or disconnected molecules, and identify the top-$k$ molecules based on generation frequency. This frequency-based ranking strategy enables direct comparison with DiffMS and other baseline methods. We note that alternative ranking strategies exist for MS-BART (Han et al., 2025), which ranks

candidates by formula distance to the target molecular formula. Here, we adopt frequency-based ranking for consistency with the established DiffMS benchmark and to ensure fair comparison across diffusion-based molecular generation methods.

## 4.3 MAIN RESULTS

**Structure Elucidation Performance.** Table 1 presents the *de novo* structure elucidation performance on NPLIB1. FlowMS achieves state-of-the-art performance on 5 out of 6 metrics, with the highest top-1 accuracy (9.15%), lowest MCES distances (9.32 for top-1, 7.96 for top-10), and highest Tanimoto similarities (0.46 for top-1, 0.51 for top-10). Compared to DiffMS, FlowMS improves top-1 accuracy by 9.7% while reducing top-1 MCES by 22% and increasing top-1 Tanimoto by 31%. These structural similarity improvements indicate that FlowMS generates chemically closer candidates even when exact reconstruction fails, providing valuable structural hints for domain experts. The top-10 accuracy of FlowMS (12.05%) does not surpass DiffMS (15.44%), which we attribute to DiffMS's longer sampling trajectory that may produce greater sample diversity. However, the superior structural similarity metrics suggest that FlowMS concentrates probability mass on high-quality candidates.

**Comparison with Baseline Paradigms.** The results in Table 1 reveal distinct performance characteristics across different approaches. Autoregressive methods Spec2Mol (Litsa et al., 2023) achieve near-zero accuracy, likely due to their inability to capture the permutation-invariant nature of molecular graphs and mass spectra. Two-stage approaches combining spectrum-to-fingerprint prediction with fingerprint-to-structure generation, MIST+MSNovelist (Stravs et al., 2022) and MIST+Neuraldecipher (Le et al., 2020), show moderate but limited performance due to error propagation. Scaffold-based generation MADGEN (Wang et al., 2025) struggles with scaffold prediction from spectra. Iterative refinement methods demonstrate substantially stronger performance. DiffMS (Bohde et al., 2025) leverages discrete graph diffusion with formula constraints, and MS-BART (Han et al., 2025) employs language model pretraining on fingerprint-molecule pairs; both achieve better performance. FlowMS inherits the iterative refinement paradigm while introducing a flow matching-based model for *de novo* molecular generation from mass spectra, achieving outstanding accuracy and structural similarity.

## 4.4 VISUALIZATION OF GENERATED MOLECULES

While *de novo* generation of the exact molecular structure remains challenging across all methods, the consistently high structural similarity achieved by FlowMS indicates practical utility for narrowing the chemical search space. As demonstrated in Figure 2, FlowMS places exact matches within the top-4 predictions and generates structurally similar alternatives when exact reconstruction fails. These close-match candidates can guide domain experts toward the correct structure by providing chemically plausible hypotheses. The visualization results of generated molecules complement the quantitative metrics in Table 1 and demonstrate the practical applicability of discrete flow matching to structure elucidation. More generated molecules results that include both FlowMS correctly identifies the target molecule within top-1 predictions, and FlowMS does not recover the exact ground truth structure in top-1 predictions, are provided in Figures 3 and 4 in the appendix.

## 5 CONCLUSION

We present FlowMS, a discrete flow matching framework for *de novo* molecular generation conditioned on tandem mass spectra. By leveraging the discrete flow matching framework for mass spectrum conditioned molecular generation, FlowMS achieves state-of-the-art performance on 5 out of 6 evaluation metrics on the widely adopted benchmark NPLIB1. The superior structural similarity scores demonstrate that FlowMS generates chemically plausible candidates, narrowing the search space for downstream expert verification. Taken together, the state-of-the-art performance and visualizations results establish discrete flow matching as a promising paradigm for structure elucidation in high-throughput metabolomics. Future research includes exploring alternative spectral encoding architectures, evaluating the scalability of FlowMS across broader spectral libraries, and leveraging the flexible design space of flow matching to investigate advanced sampling strategies.

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

# A  GENERATED MOLECULES

## A.1  NPLIB1 MOLECULES: POSITIVE EXAMPLES

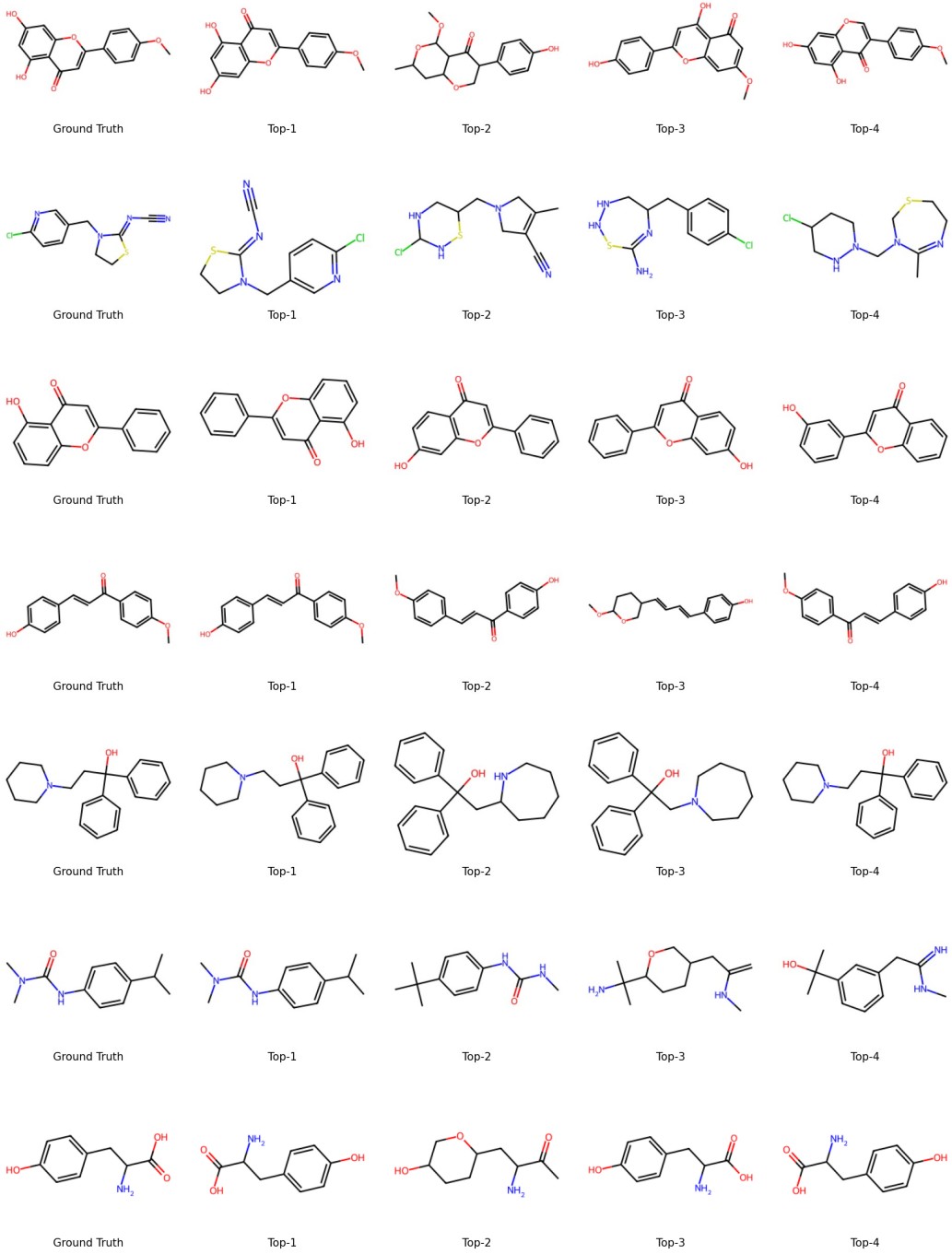

Figure 3: Positive test examples on the NPLIB1 dataset (Dührkop et al., 2021), where FlowMS correctly identifies the target molecule within top-1 predictions. Ground truth molecules (left column) and FlowMS predictions (right columns).

## A.2 NPLIB1 Molecules: Negative Examples

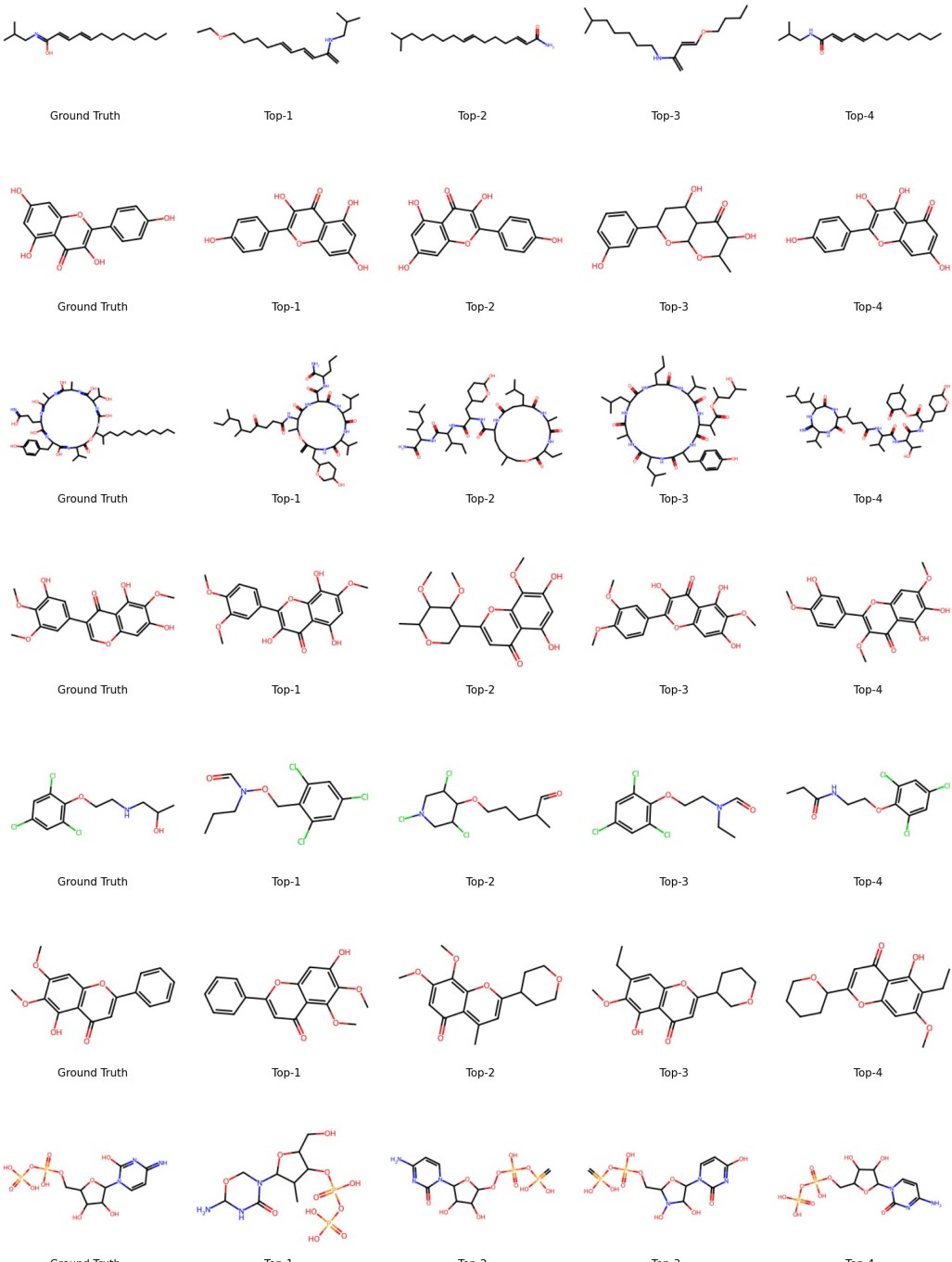

Figure 4: Negative test sample from the NPLIB1 dataset (Dührkop et al., 2021), where FlowMS does not recover the exact ground truth structure in top-1 predictions. Ground truth molecules (left column) and FlowMS predictions (right columns).

We present examples of FlowMS predictions on the NPLIB1 to illustrate the model's behavior on different cases. Figure 3 shows examples where FlowMS correctly identifies the target molecule within its top-1 prediction, demonstrating the model's ability to recover exact molecular structures from mass spectral data. Figure 4 presents failure cases where the ground truth structure does not appear in the top-1 prediction.

Notably, even in negative cases where the ground truth structure does not appear in the top-1 prediction, the predicted molecules also share significant structural features with the ground truth molecule, including similar scaffolds and functional groups. This suggests that FlowMS learns meaningful relationships between structure and spectrum for molecules across diverse molecular classes. Even when exact recovery is not achieved, the high MCES and Tanimoto scores exhibited by the representative cases in Figure 2 still reflect a high degree of structural similarity to the ground truth, indicating the model generates chemically plausible candidates in the local neighborhood of the correct structure.

## B EXPERIMENTAL DETAILS

Our implementation builds upon the publicly available codebases of DiffMS (Bohde et al., 2025) and DeFoG (Qin et al., 2024). We have generally followed the default settings for these models.

For node features $\mathbf{X}$, we use a one-hot encoding of atom types, $\mathbf{X} \in \mathbb{R}^{n \times d}$, where $d$ is the number of different atom types in the dataset. The adjacency matrix $\mathbf{A} \in \{0, 1\}^{n \times n \times b}$ represents edge types, where $b = 5$ corresponds to no bond, single, double, triple, and aromatic bonds.

For pretraining the decoder, we use 2048-bit Morgan fingerprints with radius 2 for the structural conditioning $\mathbf{y} \in \mathbb{R}^{2048}$. We train using the cross-entropy loss between the predicted edge probability distributions and the true adjacency matrix. We use the pretraining dataset provided by Bohde et al. (2025), which consists of 2.8M fingerprint-molecule pairs sampled from DSSTox (EPA, 2015), HMDB (Wishart et al., 2022), COCONUT (Sorokina et al., 2021), and MOSES (Polykovskiy et al., 2020) datasets. We initialize $p_0$ as the uniform distribution over bond types. We pretrain the decoder for 100 epochs using the AdamW optimizer (Loshchilov & Hutter, 2016) with weight decay 1e-12, and gradient clipping at 1.0. We use a cosine annealing learning rate scheduler (Loshchilov & Hutter, 2017). We pretrain the encoder on the same dataset used for finetuning, train with the multi-objective loss settings of Goldman et al. (2023b) and the RAdam optimizer (Liu et al., 2019). We finetune the end-to-end model using cross-entropy loss with no auxiliary training objectives. We use the AdamW optimizer (Loshchilov & Hutter, 2016) and cosine annealing learning rate schedule. We finetune for 50 epochs on NPLIB1 with batch size 128.

