# OpenReview forum: "FlowMS: Flow Matching for De Novo Structure Elucidation from Mass Spectra"
_ICLR.cc/2026/Workshop/FM4Science — ICLR 2026 Workshop FM4Science Poster_

### Official Review · Reviewer_dYQ9 · 2026-02-21
**Review for FlowMS: Flow Matching for De Novo Structure Elucidation from Mass Spectra**

**Rating:** 7
**Confidence:** 3

**Review:**

This is a strong applied generative modeling paper that adapts flow matching to spectrum-conditioned molecular generation and demonstrates competitive empirical performance. The paper is clearly written and technically sound, though the methodological novelty is moderate.

Pros
- De novo structure elucidation from mass spectrometry is an important and underexplored problem with clear scientific relevance.
- The paper provides a clean adaptation of discrete flow matching to conditional graph generation with formula constraints.
- The engineering design is solid: the encoder–decoder pretraining strategy is reasonable and leverages large-scale molecular pretraining corpora (~2.8M molecules).
- The empirical results are competitive, showing improvements over strong diffusion and sequence-based baselines on NPLIB1.
- The qualitative analysis is useful, with molecule visualizations demonstrating plausible structural recovery and an honest discussion of top-1 vs top-10 trade-offs compared to diffusion models.


Cons:
- The originality is limited: the core contribution largely replaces diffusion with flow matching, which reads more as a paradigm substitution than a conceptual breakthrough, similar to diffusion→flow transitions seen in other domains.
- Chemical validation is limited without domain expert assessment or downstream validation.
- The paper lacks deeper ablations; it would be helpful to include studies such as encoder freezing vs fine-tuning, sensitivity to formula constraints, or analysis of flow vs diffusion behavior

---

### Official Review · Reviewer_JXWs · 2026-02-21
**Review on FlowMS: Flow Matching for De Novo Structure Elucidation from Mass Spectra**

**Rating:** 5
**Confidence:** 4

**Review:**

# Quality

The paper adapts discrete flow matching to the spectrum-conditioned setting. Here are several quality concerns:

- The abstract and introduction motivate the work partly by noting that "diffusion-based generation for this task remains computationally demanding" (Abstract). Even though flow matching is well known, an experiment of time consumption comparison is appreciated.

- The paper does not disentangle the contributions of discrete flow matching vs. other design choices.

- Evaluation is limited to NPLIB1 only. The paper mentions MassSpecGym (Bushuiev et al., 2024) in Section 3.1 ("We exclude all molecules from the NPLIB1 and MassSpecGym from pretraining data") but does not evaluate on it.


# Clarity

The paper is generally well-written and organized. The problem formulation is clear, and the mathematical notation is consistent. Figure 1 provides a helpful overview of the framework.

# Originality

FlowMS's contribution is: (1) combining discrete flow matching with spectrum conditioning and molecular formula constraints, and (2) demonstrating empirically that this combination outperforms diffusion on NPLIB1. While this is a useful empirical finding, the methodological contribution is primarily an engineering integration of existing components (DeFoG decoder + MIST encoder + DiffMS pipeline).

# Significance

FlowMS generates chemically closer candidates even when exact reconstruction fails, providing valuable structural hints for domain experts. However, the significance is limited by three factors: (1) the narrow evaluation scope (single benchmark, no ablations, no computational analysis), (2) the smaller methodological contribution compared with other flow matching works for molecules, and (3) limited exploration into why flow matching outperforms diffusion for this specific task.

# Pros

- Table 1 shows FlowMS achieves SOTA on 5/6 metrics.

- Figure 2 provides compelling qualitative evidence that FlowMS places exact matches within the top-4 predictions and generates structurally similar alternatives when exact reconstruction fails.

- The paper follows established evaluation practices from Bushuiev et al. (2024), uses the same ranking strategy as DiffMS for fair comparison, and appropriately excludes oracle baselines.

# Cons

- The paper provides zero timing comparisons, no analysis of sampling steps, and no empirical evidence of efficiency gains.

- There is no experiment isolating the effect of flow matching vs. diffusion while controlling other factors.

- The prior $p_0(a_t^{(ij)})$ in Equation 1 is never explicitly defined in the paper.

- The paper does not discuss other works applying discrete flow matching to molecular generation or diffusion-based spectrum-conditioned generation.

- Evaluation is limited to NPLIB1.

---

### Official Review · Reviewer_NEtr · 2026-02-22

**Rating:** 6
**Confidence:** 4

**Review:**

# Summary

De novo structure elucidation from mass spectra is important in drug discovery and chemistry, but challenging due to combinatorial complexity and the fact that distinct structures can yield nearly indistinguishable fragmentation patterns. The authors propose FlowMS, a discrete flow-matching framework for generating molecular graphs conditioned on mass spectra. The model uses a pretrained MIST formula transformer to encode the spectrum into a conditioning representation. A discrete flow-matching generator then constructs the molecular graph via iterative refinement in the discrete adjacency-matrix space. The method enforces the known chemical formula as a hard constraint, focusing on heavy-atom connectivity and leaving hydrogen placement implicit.

# Strengths
* Important problem for AI-for-chemistry/drug-discovery.
* Empirical results appear strong.

# Weaknesses
* The method appears closely related to DiffMS, with the main change being replacement of the discrete diffusion generator with a flow-matching generator. However, an insight of such replacement is a benefit is limited.
* Notation could be streamlined. For example, introducing $\mathcal{M}$ seems unnecessary if the noising/denoising dynamics apply only to the adjacency variables.

# Questions
* How is $p_0(a_0^{(ij)})$ initialized? Is $p_0(a_0^{(ij)})$ uniform over bond types or set to empirical marginals from the training data (e.g., reflecting single/double bond frequencies)?

* The paper claims (line 75) that flow matching uses linear interpolation paths while diffusion has “curved noising trajectories.” This argument is not clearly justified for discrete adjacency-matrix state spaces. In discrete settings, both discrete diffusion and discrete flow-matching methods typically define CTMC jumps. Many common discrete “diffusion” corruptions also yield simple mixture/linear marginal paths in the probability simplex, so “curved trajectories” is not obviously the right distinction here.

---

### Official Review · Reviewer_vC1G · 2026-02-25
**Review of FlowMS**

**Rating:** 9
**Confidence:** 3

**Review:**

This contribution a de novo structure elucidation for molecular structure from spectra in mass spectroscopy. That is, it learns to generate the molecular structural graph of a compound (even if the structure does not previously live in a database) given its spectrum. It relies on training an encoder-decoder architecture with transformer as the encoder of the spectra and a conditional graph flow matching architecture as the decoder. This is a hard problem because the one-to-many inverse problem of going from spectra to molecular graph is not well specified (several molecular graphs can result in very similar spectra). The method achieves SOTA in 5 out of 6 benchmark tasks, which are based in retrieval (top-k accuracy) and structural similarity (MCES, Tanimoto).

Pros:

- While other diffusion-based approaches have shown promising performance, it not clear to what extent discrete flows can improve on the SOTA. This work for the first time investigates flow matching as a strategy and show promising results.

- The authors present the results with clarity, and indicate clearly where the gains of their method are with respect to existing methods.

- The description of the probabilistic denoiser is rigurous. The work combines graph neural networks with probabilistic generation.

- Baselines and bechmarks are cleary explained, and the results clearly show the numerical gains in the different metrics when the method is employed.

Cons:

- The work could have benefited from ablation studies to investigate different hyperparameters for encoder and decoder, or at least a discussion on how architectural choices are expected to affect the results.

- Enforcing a molecular formula for the target molecule fixes the nodes of the graph generation, reducing the complexity of the problem. The authors mention that the chemical formula can be determined experimentally. However, being a crucial aspect of the method, the authors should explain how feasible it is to derive the molecular formula in a realistic scenario, and how the applicability of the method relies on this determination.

---

### Decision · Program_Chairs · 2026-03-02

Accept (Poster)